# A second-order perspective on linear mode connectivity modulo permutations

**Beatrix Benkő** [*]
benkombeatrix@gmail.com

## Abstract

Linear mode connectivity of neural networks revolves around the striking observation that solutions obtained from the same initialization can often be connected by linear paths in parameter space, without substantial loss increase along them. Recent advances have demonstrated that accounting for permutations of hidden units enables linear mode connectivity even between independently trained networks. We provide insights into this phenomenon from a second-order perspective, employing local, derivative-based approximations of the loss and network outputs. We discover that such approximations systematically underestimate the rise in loss between independently trained minima, yet they can accurately capture the loss behavior along the path between the permutation-aligned solutions. We attribute the failure to higher-order variations in loss arising when interpolating misaligned units, which are eliminated by permutation reordering, while the remaining non-linear effects diminish with increasing network width. Beneath the aggregate loss values, however, we find that the network outputs and the prediction transitions they induce often evolve in a less easily traceable manner.

## 1 Introduction

Understanding the loss landscape geometry of deep neural networks [1, 2] is of fundamental importance for uncovering how they learn and perform, with broad implications for optimization and generalization. A common way to gain insight is probing the loss surface by tracing the loss along paths between different points in parameter space. By evaluating the intermediate network parameterizations obtained from interpolating the endpoints, one can study properties such as the mode connectivity of minima.

Linear mode connectivity (LMC) [3] builds around the noteworthy finding that solutions obtained via optimization from the same initialization can often be connected by linear low-loss paths in parameter space. Along such paths, the increase in loss, termed the barrier, is small: empirical values on the training and test sets remain nearly constant, with no significant drop in classification accuracy. Since its initial discovery, LMC has sparked significant interest, influencing both theoretical work [4–6] and practical methods like model merging through weight interpolation or more general arithmetic operations in parameter space [7–9].

Recent advances in this regard aim to uncover the role of parameter permutation symmetries in LMC. Entezari et al. [10] conjectured that if the permutation invariance of the networks is taken into account, SGD solutions are likely to be linearly mode-connected. This was empirically confirmed by Ainsworth et al. [11], who considered minima trained independently from different initializations, in settings of practical interest—deep convolutional and residual networks with ReLU activations, exhibiting additional symmetries. Their results have been analyzed and extended to broader settings by subsequent work [12–14], with applications to federated and class incremental learning.

---

[*]Eötvös Loránd University & HUN-REN Institute for Computer Science and Control, Budapest, Hungary

We further study the manifestation of LMC after permutation alignment through the lens of local quadratic approximations, inspecting both the loss and network output behavior along the interpolation path between minima. For independently trained SGD solutions, such approximations systematically underestimate loss barriers as model width increases, falsely indicating LMC where it does not actually hold. We then demonstrate that the exactness of the approximations improves markedly when interpolating between permutation-aligned solutions. This improvement arises from vanishing higher-order loss derivatives along the altered path as a consequence of minimizing the $L_2$ distance between the endpoints via weight matching, not merely from the parameter distance reduction due to permutation. We showcase that similar approximations of network outputs can capture their changes, but their exactness diminishes away from the endpoints, and varies across inputs.

## 2 Background

**Notations**    We define a classifier neural network with learnable parameters stacked into a vector $\theta \in \Theta \subseteq \mathbb{R}^d$ and taking inputs $x \in \mathcal{X}$, as $f_\theta : \Theta \times \mathcal{X} \to \mathbb{R}^C$, with $C$ denoting the number of classes. The network output for an input $x$, denoted $f_\theta(x) = z(\theta)$, is a vector of unnormalized scores, referred to as logits. For training and evaluation purposes, we use the standard cross-entropy classification loss, defined as $\ell : \mathbb{R}^C \times \mathcal{Y} \to \mathbb{R}$, $\ell(f_\theta(x), y) = -e_y^T \log \operatorname{softmax}(f_\theta(x))$, where $e_y \in \{0, 1\}^C$ is the one-hot encoding of label $y \in \mathcal{Y}$. The empirical loss over a set of $N$ samples is given by: $L(\theta) = \dfrac{1}{N} \sum_{(x,y)} \ell(f_\theta(x), y)$, computed in practice as the average over mini-batches.

We consider model pairs with identical architectures, each converging to a local minimum. They are obtained by initializing two independent parameter vectors with different random seeds, and training them separately until convergence, with the same variant of SGD and identical hyperparameters, but distinct random seeds for mini-batch sampling and data augmentation. We identify the resulting models with their learnable parameter vectors, e.g., $\theta_A$ and $\theta_B$.

**Linear weight interpolation and LMC**    The linear interpolation between two parameter vectors, $\theta_A$ and $\theta_B$, denoted as $\theta_A \to \theta_B$, results in convex combinations $\theta_\lambda$ that lie along the line segment connecting them in parameter space:    $\theta_\lambda = (1 - \lambda)\,\theta_A + \lambda\,\theta_B$. The coefficient $\lambda \in [0, 1]$ controls the relative contribution of the two models. We use the notation $\theta_\lambda$ for interpolated parameters, even when $\theta_B$ is replaced with a permuted version of it. We refer to the loss of $\theta_\lambda$ models as interpolation loss. The LMC property between minima holds when the interpolation loss does not increase significantly, as measured by the barrier—the maximum increase in it along the linear path, relative to the linearly interpolated loss values at the endpoints.

**Permutations of parameters**    Changing the order of units (filters and neurons) within any hidden layer of a network does not alter its outputs, resulting in functionally equivalent parameterizations. Due to such permutation symmetries of parameters, independently trained models may converge to parameterizations with similar hidden units arranged in different orders within layers. Naively interpolating the misaligned units knowingly degrades the performance of intermediate models $\theta_\lambda$ and inflates the loss barrier [11]. To mitigate the issue, one model can be aligned to another via appropriate permutation transformation of all parameters. Finding suitable permutation is non-trivial, and several heuristic algorithms have been proposed. These include weight matching—i.e., minimizing the $L_2$ distance between parameters without access to data—, directly learning the mapping [11], or approximating it via a differentiable Sinkhorn operator [12]. In our experiments, we employ the weight matching algorithm from [11] to align $\theta_B$ to $\theta_A$, with $\pi(\theta_B)$ denoting its permuted counterpart.

## 3 Experimental results

We train progressively widened ResNet-20 models on the CIFAR-10 dataset, to obtain parameter vectors $\theta_A$ and $\theta_B$ as described. The base model width is set so that the first convolutional layer has 16 filters. We increase the model width by scaling the number of neurons or convolution filters in each layer by factors of 1, 2, 8, and 32. We also adopt the training configurations from [11].

### 3.1 Loss behavior and its approximations

We approximate the interpolation loss $L(\theta_\lambda)$ as a quadratic function of the parameters centered at the endpoint $\theta_A$ as:

$$L(\theta_\lambda) \approx L(\theta_A) + (\theta_\lambda - \theta_A)^\top \nabla_\theta L(\theta_A) + \frac{1}{2}(\theta_\lambda - \theta_A)^T \nabla_\theta^2 L(\theta_A)(\theta_\lambda - \theta_A) + \mathcal{E}_2(\theta; \theta_A),$$

where $\nabla_\theta L(\theta_A)$ denotes the gradient vector of the loss evaluated at $\theta_A$, $\nabla_\theta^2 L(\theta_A) = \nabla_\theta (\nabla_\theta L(\theta_A))$ denotes the Hessian matrix evaluated at $\theta_A$. For the networks with ReLU activations considered here, we define the derivatives based on the update rule used in practice. The approximation of $L(\theta_\lambda)$ centered at another solution $\theta_B$ or its permuted version $\pi(\theta_B)$ can be obtained in similar vein. The linear approximations result from retaining the first two terms of the equations. We compute the first-order terms using the built-in gradient method, and the second-order terms using Hessian–vector products via automatic differentiation, evaluated with batch size of 50.

Figure 1 shows how closely these approximations track the test loss (up to the midpoint). The interpolation loss is computed at 25 evenly spaced points along the linear path between $\theta_A$ and $\theta_B$. While the approximations are accurate in the proximity of the endpoints, even the quadratic ones incur substantial error as the network width increases, failing to account for the rise in the loss.

In contrast, Figure 2 demonstrates that second-order approximations remain highly accurate up to the midpoint when interpolating between $\theta_A$ and the permutation-aligned $\pi(\theta_B)$ (with $\pi$ obtained via weight matching). Notably, this holds for all considered widths, not just for the one where near-zero barrier has been achieved in [11]. Meanwhile, linear approximations track the loss only in the very close vicinity of the endpoints, reflecting their inability to describe curvature. We observe a similar trend of improved quadratic approximations also with MLP models trained on MNIST and ResNet-20 models trained on CIFAR-100, albeit with less precise fit in some cases (see Figures 3 to 6).

We attribute the large error toward $\theta_B$ to pronounced nonlocal, higher-order variations of the interpolation loss that go beyond what local derivatives can capture, rather than merely to the larger distance between endpoints. These variations stem from interpolating misaligned units, which induces abrupt changes in the decision boundaries and collapses model functionality (as illustrated in Figure 7)—leading to accuracy that quickly degrades to random guessing.

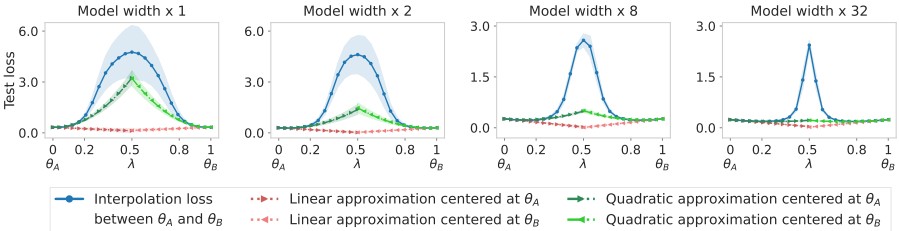

Figure 1: Interpolation loss and its linear and quadratic approximations centered at $\theta_A$ and $\theta_B$, corresponding to fully trained ResNet-20 models on CIFAR-10. Width denotes the multiplicative factor by which the number of filters and neurons is increased in every hidden layer. Lines represent mean values across ten distinct model pairs, shaded areas indicate $\pm$ standard deviation around them.

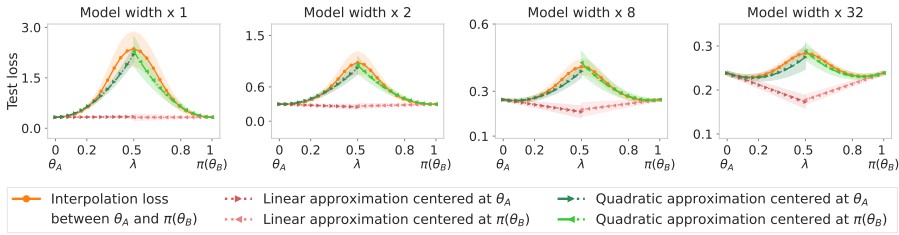

Figure 2: Interpolation loss and its linear and quadratic approximations centered at $\theta_A$ and $\pi(\theta_B)$. $\pi$ is obtained via weight matching $\theta_B$ to $\theta_A$. Other details are as in Figure 1. Quadratic approximations closely track the loss, reflecting the curvature present even between the $32\times$-wide models.

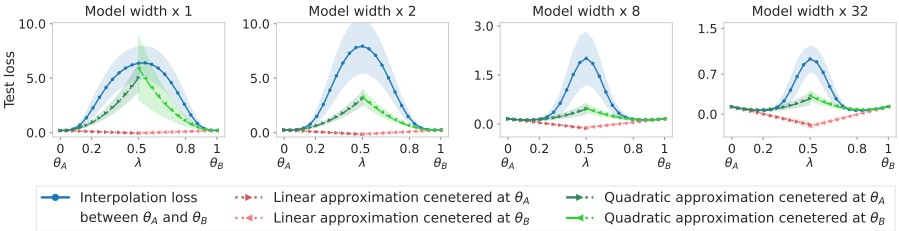

Figure 3: Interpolation loss and its linear and quadratic approximations centered at $\theta_A$ and $\theta_B$, corresponding to fully trained MLP models on MNIST. Other details are as in Figure 1.

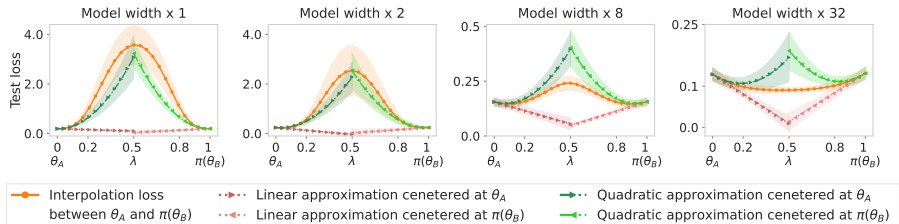

Figure 4: Interpolation loss and its linear and quadratic approximations centered at $\theta_A$ and $\pi(\theta_B)$. $\pi$ is obtained via weight matching $\theta_B$ to $\theta_A$. Other details are as in Figure 3.

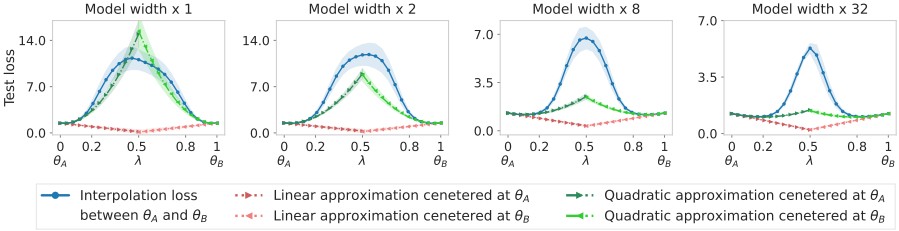

Figure 5: Interpolation loss and its linear and quadratic approximations centered at $\theta_A$ and $\theta_B$, corresponding to fully trained ResNet-20 models on CIFAR-100. Other details are as in Figure 1.

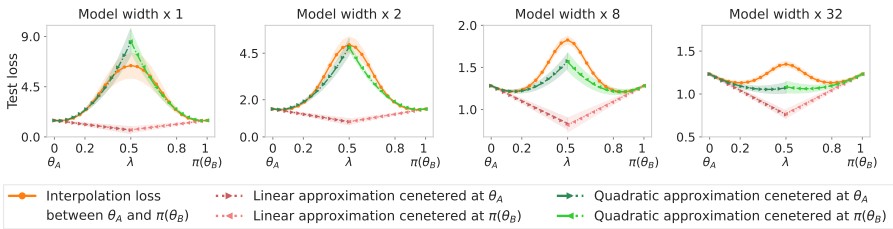

Figure 6: Interpolation loss and its linear and quadratic approximations centered at $\theta_A$ and $\pi(\theta_B)$. $\pi$ is obtained via weight matching $\theta_B$ to $\theta_A$. Other details are as in Figure 5.

### 3.2 Logit-level dynamics and its approximations

Focusing on the $32\times$-wide model pairs, where network outputs should be relatively stable, we ask whether their variations are captured by local approximations centered at the endpoints. We again employ local, derivative-based approximations, this time at the logit level, specifically for the $k$-th component of the logit vector $z(\theta_\lambda)$ around the endpoint $\theta_A$ given by:

$$[z(\theta_\lambda)]_k \approx [z(\theta_A)]_k + (\theta_\lambda - \theta_A)^\top \nabla_\theta [z(\theta_A)]_k + \frac{1}{2}(\theta_\lambda - \theta_A)^\top \nabla_\theta^2 [z(\theta_A)]_k (\theta_\lambda - \theta_A).$$

The first two terms yield a linear approximation based on the gradient of the $k$-th logit at $\theta_A$, corresponding to a row of the logit Jacobian. Analogous approximations centered at $\theta_B$ and $\pi(\theta_B)$ can be obtained by replacing $\theta_A$ with the respective endpoints and using the appropriate parameter

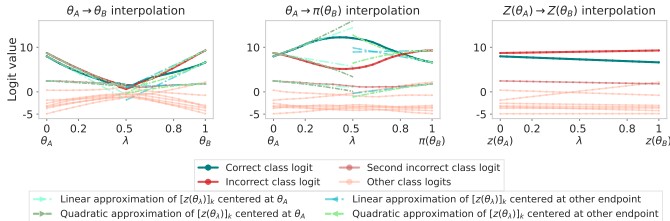
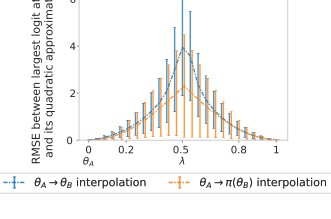

Figure 7: Comparison of how the logits of a single input change when interpolating parameters from $\theta_A$ to $\theta_B$ and $\pi(\theta_B)$, evaluating the $z(\theta_\lambda)$ logits at 25 step, as well as when directly interpolating the logits $z(\theta_A)$ and $z(\theta_B)$. The sample is misclassified by both $\theta_A$ and $\theta_B$, weight interpolation leads to correct prediction at some steps.

Figure 8: RMSE between the $k$-th logit in $z(\theta_\lambda)$ and its approximations, with $k$ as the index of largest logit in $z(\theta_A)$ per sample. Lines show mean over all test samples; error bars: $\pm$ SD.

interpolation formula to express $\theta_\lambda$. We computed these in similar way as loss approximations, separately for each output dimension.

Figure 7 illustrates how the 10-dimensional logits $z(\theta_\lambda)$ of a single input vary during interpolation from $\theta_A$ to $\theta_B$ and $\pi(\theta_B)$ in parameter space, as well as during direct linear interpolation of the endpoint logits, i.e., $(1 - \lambda)\, z(\theta_A) + \lambda\, z(\theta_B)$. In addition, it shows the linear and quadratic approximations centered at endpoints for the three largest-magnitude components of $z(\theta_\lambda)$. Along the path to $\theta_B$, the logits collapse to near-zero values, and $\theta_\lambda$ loses its ability to classify inputs—an issue known as variance collapse [13]. In contrast, along the path to $\pi(\theta_B)$, the quadratic effect of weight interpolation corrects the wrong class prediction of the endpoints with high confidence. Direct interpolation of the logits just blends the incorrect outputs from $\theta_A$ and $\theta_B$, without correction, highlighting the advantage of parameter interpolation over simple output ensembling.

This example demonstrates a more general trend: for many inputs, the considered linear approximations capture the logit-level dynamics near endpoints, while quadratic ones can better track it in some cases. Yet a high root mean squared error of quadratic approximations toward the midpoint along both paths in Figure 8 tells that there remain samples for which $z(\theta_\lambda)$ varies beyond what these approximations can describe. With narrower networks, logit variations tend to be even more irregular—as we observed by visualizing the logit interpolation with $8\times$-, $2\times$-, and $1\times$-wide models for the same input images.

## 4 Conclusions

We investigate whether local quadratic approximations capture the loss behavior along interpolation paths between independently trained SGD solutions. We discover that they underestimate loss barriers when increasing model width, falsely indicating LMC, but the main source of error is eliminated by permutation alignment of endpoints, with higher-order loss derivatives vanishing along the altered path. Beneath the aggregate loss values, we find that prediction transitions are governed by structured logit evolution determined by the interpolation direction. Even for extremely wide networks, local linear and quadratic approximations of logits capture the network output dynamics in the close proximity of endpoints only. While this can be leveraged to explain related empirical observations [15], deviations away from endpoints and at the sample level require further inquiry—we hypothesize that this gap could not be significantly reduced even with higher-order local approximations, because of their limited validity. Accordingly, the LMC observed predominantly in computer vision can be attributed to structured, direction-dependent changes in network outputs on the usually considered image datasets.

## Acknowledgments

This work was supported by the European Union project RRF-2.3.1-21-2022-00004 within the Artificial Intelligence National Laboratory (MILAB).

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
