# OpenReview forum: "A second-order perspective on linear mode connectivity modulo permutations"
_NeurIPS.cc/2025/Workshop/UniReps — UniReps2025_

### Official Review · Reviewer_FanG · 2025-09-03
**Review of A Second-Order Perspective on Linear Mode Connectivity Modulo Permutations**

**Confidence:** 3

**Review:**

The paper studies linear mode connectivity (LMC) between neural network solutions from a second-order perspective. Using local quadratic approximations of the loss and logits, the authors analyze interpolation paths between independently trained models and their permutation-aligned counterparts. They show that quadratic approximations systematically underestimate the loss barrier between independently trained solutions, but become accurate after permutation alignment, since higher-order loss variations vanish. Experiments with ResNet-20 models of varying width on CIFAR-10 and CIFAR-100, as well as MLPs on MNIST, illustrate these effects. At the output level, the work also shows that prediction transitions follow structured but complex dynamics, which quadratic approximations can only capture close to the endpoints.

Strengths

- Provides a clear empirical analysis of why permutation alignment enables low-loss linear connectivity.
- Uses systematic quadratic approximations of loss and logits, offering insight into the role of higher-order variations.
- Experiments cover multiple settings (different widths, datasets, and architectures), reinforcing the observations.
- Figures and explanations are detailed and make the results accessible.
- The conclusions help explain previously observed behavior of mode connectivity.

Weaknesses

- Experiments are limited to relatively small-scale models (ResNet-20, MLPs). It is unclear how the findings extend to larger architectures.
- While the analysis shows where quadratic approximations succeed or fail, it does not suggest improvements beyond alignment.
- Output-level approximations show irregularities that are acknowledged but not deeply analyzed.

Suggestions for Authors

- Clarify early that the main contribution is analysis rather than methodological novelty.
- Consider extending the study to larger-scale architectures to test the generality of the findings.
- Provide further discussion on the implications of the observed output dynamics for understanding generalization and connectivity.
- Explore whether refined approximations could reduce the gap between theory and observed logit behavior away from endpoints.

Overall, this paper provides a clear investigation of second-order approximations in linear mode connectivity. While it does not introduce new methods, it offers useful insights into the role of permutation alignment and the limits of local approximations, making it a suitable contribution for the UniReps workshop.

**Score:**

3

**Topic Fit:**

3

---

### Official Review · Reviewer_texK · 2025-09-11
**Review of "A second-order perspective on linear mode connectivity modulo permutations"**

**Confidence:** 3

**Review:**

## Summary:
The paper offers preliminary insights into linear mode connectivity (LMC) between independently trained models when permutations of filters and neurons are considered. The authors analyze LMC after performing permutation alignment, employing local quadratic approximations to study the loss landscape. Their initial results show that while these approximations capture the loss behavior along interpolation paths, they underestimate loss barriers as model width grows, which can give a false impression of LMC. These observations are supported by two well-designed experiments.

## Strengths
- Clear introduction of the problem and positioning of the extended abstract within the existing literature.
- Methodology used in the paper is clear and well explained. The authors explained most of their decisions.

## Weaknesses:
- A few questions stayed on my mind while reading the paper:
1 - There is a slight difference between the interpolation loss and its quadratic approximations. How much of this difference is inherited from  the errors introduced by the alignment algorithm [11] and how much of this is the error of the chosen approximation method?
2 - It is not clear how the authors got to the statement "The logit variations tend to be even more irregular" (line 135) since they only focused on the 32x-wide model pairs. A bit more explanation of this statement would be beneficial for readers.

## Recommendation:
I recommend this paper to be accepted. The extended abstract brings forward an interesting insight that fits the workshop's themes, and it would benefit the attendees of the workshop to look at LMC from a different perspective.

**Score:**

4

**Topic Fit:**

2

---

### Official Review · Reviewer_r1LE · 2025-09-16
**Probing LMC with alignment and quadratic fits**

**Confidence:** 3

**Review:**

**Quality:** Carefully executed empirical study. The authors compute true interpolation loss between independently trained nets and compare it to first- and second-order Taylor approximations at the endpoints. Before alignment, even the quadratic fit underestimates the barrier (worse with width); after permutation alignment (weight matching), the quadratic tracks the true curve well up to the midpoint. Methods are sound and reproducible. Main limitation is scope (ResNet-20/CIFAR-10).

**Clarity:** From what I understood, this paper gives a mechanical understanding of linear mode connectivity in permutation invariance. Since second-order approximations holds better when interpolating between $ \theta_1 $ and $ \pi(\theta_2) $, this means that not only the barrier is lower, but also the loss landscape between these two parameter settings is smoother.

In line 43 and 44, the authors mention “minimizing the L2 distance between the endpoints” but only explain what they actually mean later: weight matching (in line 75).

Figure 3 and 4 is way too close to each other, hard to read. I also found understanding Figure 3 itself difficult.

**Originality:** Adds a useful, mechanism-level angle. Misalignment introduces higher-order, non-local effects along the path, while weight matching pairs like-for-like units so higher-order terms shrink. As a result, second-order fits agree with the measured interpolation loss up to mid-path. This mechanistic explanation, to my knowledge, is a fresh contribution.

**Significance:** Offers a practical tool for checking the LMC, but before blindly relying on second-order approximations of the interpolation loss, more experiments should be carried with bigger networks and datasets.

**Pros:**
- Results are consistent across width sweeps, not just a single scale.

**Cons:**
- Narrow experimental scope (ResNet-20/CIFAR-10; limited architectures/datasets).
- Relies on one alignment heuristic; little comparison to alternatives.

**Score:**

3

**Topic Fit:**

2

---

### Official Review · Reviewer_dgbU · 2025-09-16
**Review of "A second-order perspective on linear mode connectivity modulo permutations"**

**Confidence:** 4

**Review:**

The authors provide numerical results and justifications explaining linear mode connectivity (modulo permutations) in networks trained from different seeds. The paper is very easy to follow and fits the theme of the workshop.

Strengths:
1) This paper is well-written, is easy to follow, and fits the theme of the workshop. The introduction section does a great job of orienting with previous literature.

2) The explanations make sense.


Weaknesses/suggestions:
1) Some permutation-based or related alignment metrics such as semi-matching and soft-matching would be good to include in the discussions.

2) Figure 3 and 4 placement currently makes the figure captions crowded.

3) I would be good to see how these results generalize to larger datasets.

4)  (Minor point): The openreview submission had a misspelling: "A second-order perspective on liear mdoe connctivity modulo permutations". I did consider the possibility that 'mode' might have been intentionally misspelled as 'mdoe' as a permutation joke but the title in the pdf says 'mode' as compared to 'mdoe' in the openreview submission so I am not sure. While this does not affect my recommendation to accept this paper, If this was a mistake instead of a joke, I would highly recommend the authors to spend more time proofreading since some reviewers would deem this unacceptable.

**Score:**

4

**Topic Fit:**

3